# Integrated Analysis of Metabolome and Transcriptome Provides Insights into Flavonoid Biosynthesis of Pear Flesh (*Pyrus pyrifolia*)

**DOI:** 10.3390/foods14213716

**Published:** 2025-10-30

**Authors:** Jun Su, Yanping Liang, Yingyun He, Wen Zhang, Jingyuan Zhou, Lina Wang, Songling Bai

**Affiliations:** 1College of Agriculture and Biotechnology, Zhejiang University, Hangzhou 310058, China; sujun198067@163.com; 2Horticulture Research Institute, Yunnan Academy of Agricultural Sciences, Kunming 650205, China; liangyanping2002@126.com (Y.L.); hyy2050@163.com (Y.H.); 18188468785@163.com (W.Z.); 3Faculty of Food, Drugs and Health, Yunnan Vocational and Technical College of Agriculture, Kunming 650205, China; 18088296527@163.com

**Keywords:** pear flesh, flavonoids, transcriptome, metabolome, *MYB4*

## Abstract

The flavonoids in the flesh significantly impact fruit quality and nutritional value. In this study, the flesh of ‘Heqingxiaoshali’ (HF) and ‘Lunanhuangpingli’ (LF) was analyzed by non-targeted metabolomics and transcriptomics. The results showed that the contents of reducing sugars, titratable acids and total flavonoids in HF flesh were significantly higher than those in LF. Metabolomics analysis revealed significant differences in lipids, organic acids, phenylpropanoids, and polyketides between HF and LF at each developmental stage, with Trilobatin, Cratenacin, and Betuletol 3-galactoside showing significant differences across all stages, and proanthocyanidins being the most abundant flavonoids in HF at harvest. Transcriptome analysis revealed significant differences in genes related to flavonoid biosynthesis between the two varieties, with differentially expressed genes enriched in the “phenylpropanoid biosynthesis” and “flavonoid biosynthesis” pathways across at least four developmental stages. WGCNA suggested that differences in the flavonoid accumulation were closely related to seven structural genes (*PAL*, *CHI*, *FHT*, *FLS*, *DFR*, *ANS* and *ANR*) and a transcription factor (*MYB4*), as well as genes related to auxin response and jasmonic acid metabolism. This study provides new insights into the molecular mechanisms of flavonoid accumulation in the fruit flesh of pears and offers a theoretical basis for pear fruit quality improvement.

## 1. Introduction

Pear (*Pyrus* sp.) is a major fruit crop in temperate regions and is the second most widely cultivated fruit tree in the world [1]. Flavonoids, a class of secondary metabolites secreted by plants, contribute to the color of pear fruit and response to abiotic stress and affect fruit quality [2,3]. Yunnan, as one of the central regions where the genus Pyrus originated, has rich resources of niche varieties [4]. These varieties typically possess a long history and strong adaptability to the local environment, which has led to the development of unique qualities. Their metabolic components, such as the content and composition of flavonoids, often exhibit significant differences [5]. Using these resources to conduct an in-depth investigation into flavonoid composition and the molecular mechanisms underlying the differences between varieties can facilitate a comprehensive understanding of their flavor characteristics and nutritional value, thereby providing valuable references for breeding programs.

There are many kinds of flavonoid compounds in pear fruits, including anthocyanins, flavonols, catechins, polymeric flavan 3-ols (proanthocyanidins, syn. condensed tannins), and flavanones [2]. The flavonoid composition and content differ among pears of different varieties and developmental stages. A study investigated the distribution of triterpenes and phenolic compounds in different parts of ‘Ladana’ pear trees, revealing that pear peel has the highest triterpene concentration, while leaves exhibit the highest phenolic content and antioxidant capacities [6]. Another study conducted a comparative analysis of 10 common pear cultivars grown in Korea (four *Pyrus* spp.) and revealed that Niitaka and Hanareum pears possessed significantly higher levels of total phenolics, arbutin, and chlorogenic acid than other cultivars, thereby emphasizing their potential as valuable sources of bioactive compounds [7]. Additionally, studies on the physiological characteristics and sugar metabolism of various Asian pear cultivars have revealed that while the total antioxidant activity of pear fruit reaches 70% at 70 days after flowering, this activity declines sharply to approximately 36% as the fruit nears horticultural maturity [5].

The biosynthesis and metabolism of flavonoids in pears are governed by intricate gene regulatory networks that significantly influence peel coloration through anthocyanin metabolism. Light exposure has been shown to regulate anthocyanin accumulation: as studies have demonstrated that in ‘Mantianhong’ pears, *PybZIPa* activates the *PyUFGT* gene promoter to enhance anthocyanin synthesis [8], while in the red Chinese sand pear cultivar ‘Yunhongyihao,’ the down-regulation of *PyPIF5* and overexpression of *PymiR156a* boost anthocyanin levels via the *PyPIF5*-*PymiR156a*-*PySPL9*-*PyMYB114*/*MYB10* pathway [9]. Research on the green Chinese pear ‘Zaosu’ and its red mutant ‘Hongzaosu’ revealed that the accumulation of β-galactoside in the red mutant stimulates the biosynthesis of other flavonoids, leading to distinct biosynthetic profiles between the two cultivars [10]. In the red pear cultivar ‘Starkrimson’ and its green mutant, *AP2* and *WARK* transcription factors regulate anthocyanin biosynthesis, whereas *ANR* and *LAR* promote proanthocyanidin biosynthesis [11]. Additionally, the flavonoid 3′-hydroxylase (*F3’H*) gene in Chinese pear has been shown to affect skin color by modulating anthocyanin accumulation [12]. Moreover, temperature significantly influences the coloration of pear fruit peels. Under low-temperature stress, mRNA and metabolome sequencing of floral organs in *Pyrus hopeiensis* revealed a series of differentially expressed genes (DEGs) and metabolite changes associated with the response to low-temperature stress, elucidating the biosynthesis mechanism of flavonoids [13].

In pears, research on flavonoid accumulation in fruit flesh is far less extensive than that in the peel. As previously reported, the flavonoids in pear fruit are predominantly composed of B-ring dihydroxyflavonol derivatives, such as quercetin and isorhamnetin, as well as monomeric and polymeric flavan-3-ols, such as catechin and proanthocyanidins [2]. To understand the genetic regulation of flavonoid biosynthesis in pear fresh, a recent study based on the transcriptome analysis of red-fleshed and white-fleshed pears identified an ethylene response factor (ERF), *PcERF5*, from *Pyrus communis*. The expression of *PcERF5* in fruit flesh is significantly correlated with anthocyanin content, and it activates the transcription of flavonoid biosynthesis genes (*PcDFR*, *PcANS*, and *PcUFGT*) and two key transcription factors, *PcMYB10* and *PcMYB114*, while also interacting with PcMYB10 to form an ERF5-MYB10 protein complex that enhances its transcriptional activation of target genes [14].

Flavonoids in pear fruits are an important class of plant secondary metabolites. In recent years, research on the metabolic mechanisms of flavonoids has grown, with a primary focus on anthocyanins in fruit peel and their associated coloration processes. However, studies on other types of flavonoids are relatively limited, and even less attention has been paid to flavonoids in the fruit flesh. Given that flavonoids in the flesh are directly related to the nutritional value of the fruit, this study builds on previous research by selecting two local varieties from Yunnan with significant differences in total flavonoid content. Using a combination of metabolomic and transcriptomic analyses across different developmental stages, we aim to explore the metabolic mechanisms of flavonoids in pear flesh. This research is expected to provide a theoretical basis for future studies on fruit quality improvement and the development of pear cultivars with enhanced nutritional profiles.

## 2. Materials and Methods

### 2.1. Plant Materials

The experimental varieties were ‘Heqingxiaoshali’ (HF) pear (*Pyrus pyrifolia*) and ‘Lunanhuangpingli’ (LF) pear (*Pyrus pyrifolia*), which were planted at the experimental base of the Yunnan Academy of Agricultural Sciences in Kunming, China (25.3549° N, 103.1152° E). The experiment was conducted from March to August 2024, with sampling times at 50, 80, 110, 140, and 170 days after flowering (DAFs). Ten fruits of uniform size were collected each time and randomly divided into three groups, with three biological replicates in each group. After peeling, the flesh was mixed and placed in cryotubes. Sampling was carried out on a clear morning. After the samples for all time periods were collected, they were tested uniformly. The samples were immediately stored in liquid nitrogen after collection and then stored in a −80 °C refrigerator after returning to the laboratory.

### 2.2. Determination of Primary and Secondary Metabolites

The levels of reducing sugars, total flavonoids, and tannins in each sample were evaluated using kits from Grace Biotechnology Co., Ltd., Suzhou, China. The reducing sugar content was determined using a total sugar content kit (G0503W), the total flavonoid content was measured with a flavonoid content kit (G0118W), and the tannin content was assessed using a tannin content kit (G0118W). The assays were performed following the manufacturer’s instructions. The titratable acid content was determined by acid-base titration, following the procedure outlined in the national standard GB 12456-2021/XG1-2025 [15]. Phenolphthalein was used as an indicator, and the endpoint was determined by the color change. The acid content was calculated based on the volume of standard sodium hydroxide solution consumed and expressed as citric acid (g/L). Three biological replicates were analyzed for each variety to ensure the reliability and reproducibility of the results.

### 2.3. Metabolite Sequencing and Analysis

In this study, liquid chromatography-mass spectrometry (LC-MS) was used to detect and analyze the metabolites of 30 samples from two varieties across five different periods. LC-MS/MS analyses were performed using a Vanquish UHPLC system (ThermoFisher, Darmstadt, Germany) coupled with an Orbitrap Q ExactiveTM HF mass spectrometer or Orbitrap Q ExactiveTMHF-X mass spectrometer (Thermo Fisher, Darmstadt, Germany) in Novogene Co., Ltd. (Beijing, China). Samples were injected into a Hypersil Goldcolumn (100 × 2.1 mm, 1.9 μm) using a 12 min linear gradient at a flow rate of 0.2 mL/min. The eluents for the positive and negative polarity modes were eluent A (0.1% FA in Water) and eluent B (Methanol). The solvent gradient was set as follows: 2% B, 1.5 min; 2–85% B, 3 min; 85–100% B, 10 min; 100–2% B, 10.1 min; 2% B, 12 min. Q ExactiveTM HF mass spectrometer was operated in positive/negative polarity mode with spray voltage of 3.5 kV, capillary temperature of 320 °C, sheath gas flow rate of 35 psi and aux gas flow rate of 10 L/min, S-lens RF level of 60, Aux gas heater temperature of 350 °C.

The raw data obtained were preprocessed using XCMS software (v 4.0), and metabolites with a coefficient of variance (CV) less than 30% in the quality control (QC) samples were retained as the final identification results. Metabolite identification was performed using a secondary spectrum library based on standards as well as secondary spectra from multiple public databases, including HMDB, RefMetaDB, and ReSpect for Phytochemicals. The reliability of identification is categorized into five levels, ranging from level 0 to level 4, with higher numbers indicating lower reliability [16]. In this study, among the detected metabolites, there were 290 at level 1, 859 at level 2, and 648 at level 3. XCMS software was also used to integrate the chromatographic peaks detected in the samples, with the peak area of each characteristic peak representing the relative quantification of the corresponding metabolite. The quantification results were normalized using the total peak area to obtain the final quantitative values of the metabolites, which were then classified. Based on the above results, data quality control, principal component analysis (PCA), and partial least squares discriminant analysis (PLS-DA) were performed (Appendix A). Differentially expressed metabolites (DEMs) were screened based on a variable importance in projection (VIP) value greater than 1 for the first principal component of the PLS-DA model, a fold change (FC) greater than 1.5 or less than 0.667, and a *p*-value less than 0.05. Subsequently, the DEMs were mapped to the KEGG database, and their significance was determined by a hypergeometric test *p*-value (*p* < 0.05).

Regarding the quantitative detection of flavonoid monomers, standard solutions of different concentrations were prepared, namely 0.5 nmol/L, 1 nmol/L, 5 nmol/L, 10 nmol/L, 20 nmol/L, 50 nmol/L, 100 nmol/L, 200 nmol/L, 500 nmol/L, 1000 nmol/L, 2000 nmol/L, 5000 nmol/L, 10,000 nmol/L, and 20,000 nmol/L. The corresponding chromatographic peak intensity data for the quantitative signals of each concentration standard were obtained. Standard curves for different substances were plotted by using the concentration ratio of external standard to internal standard or the concentration of external standard (Concentration Ratio or Concentration) as the *x*-axis, and the peak area ratio of external standard to internal standard or the peak area of external standard (Area Ratio or Area) as the *y*-axis. The standard curve equations and correlation coefficients for each flavonoid monomer are presented in Appendix A. Finally, the integrated peak area ratios of all detected samples were substituted into the linear equations of the standard curves for calculation. After further substitution into the calculation formulas, the content data of the flavonoid substances in the actual samples were ultimately obtained.

### 2.4. RNA Sequencing and Data Analysis

RNA-seq sequencing was primarily conducted on the 30 samples using the Illumina sequencing platform. Initially, the total RNA of the samples was extracted using the modified CTAB method and its integrity was analyzed using an Agilent 2100 bioanalyzer (Agilent, Palo Alto, CA, USA). Following quality control validation, libraries were constructed. After verification that library construction met quality standards, different libraries were pooled based on the required effective concentration and target sequencing data volume and subsequently sequenced on the Illumina platform. Raw data were filtered, and quality control steps (including sequencing error rate assessment and GC content distribution analysis) were performed to generate clean reads for subsequent analyses. Clean reads were then aligned to the reference genome using HISAT2 software (v 2.0.5). The reference genome was ‘Yunhong NO.1’ (*Pyrus pyrifolia*) (http://pyrusgdb.sdau.edu.cn/download_data.html, accessed on 21 October 2024). After alignment, annotations were performed using databases such as Pfam, SUPERFAMILY, GO, and KEGG. Differentially expressed gene (DEG) analysis was performed using DESeq2 software (v 1.20.0), with the screening criteria for DEGs set as |log_2_(FoldChange)| ≧ 1 and padj ≦ 0.05.

### 2.5. Weighted Gene Co-Expression Network Analysis

A weighted gene co-expression network was constructed using the WGCNA R package (v1.4.1717) with default parameters to analyze the correlation between DEG expression levels and DEM expression profiles [17]. The FPKM values for genes and the peak values of metabolites were used as matrices for Pearson’s partial correlation analysis. The top ten flavonoids with the most significant differences between the two varieties at 110 DAFs were used as phenotypic data to identify related gene expression modules. Significant trait-related modules were identified based on high correlation values. Using default settings, Hub genes from the MEred module were exported for Cytoscape software (v.3.10.3) [18].

### 2.6. RT-qPCR Analysis

Ten genes were selected for RT-qPCR analysis to validate the transcriptome results. Histone genes were employed as internal references, and gene expression levels were normalized using the 2^−∆∆Ct^ method. The specific primers used in this study are listed in Appendix A. First-strand cDNA synthesis was performed using the Evo M-MLV RT Mix Kit with gDNA Clean for qPCR Ver.2 (Accurate, Tianjin, China). RT-qPCR reactions were carried out using the SYBR qPCR Master Mix (Accurate, Tianjin, China) on an ABI QuantStudio 6 Flex Real-Time PCR System (Carlsbad, CA, USA).

## 3. Results

### 3.1. Metabolite Accumulation in the Flesh of Two Pear Varieties

Although ‘Heqingxiaoshali’(HF) and ‘Lunanhuangpingli’(LF) shared similar appearances (Figure 1A), the accumulation patterns of primary and secondary metabolites in the two varieties exhibited significant differences during fruit development. Specifically, the reducing sugar content in HF exhibited two peak values at 80 days after flowering (DAFs) and 140 DAFs, respectively, yet displayed an overall downward trend. In contrast, the reducing sugar content in LF decreased continuously throughout fruit development and remained lower than that in HF during the entire developmental period (Figure 1B). The contents of titratable acid and total flavonoids in the HF were significantly higher than those in the LF, although the trends of change were different. The contents of titratable acid and total flavonoids in the HF initially increased and then decreased with fruit development, whereas those in the LF continuously decreased throughout the entire fruit development period (Figure 1C,D). The tannin content in both cultivars decreased as the fruit developed, with the LF cultivar exhibiting a higher initial content than the HF cultivar but a lower final content at maturity (Figure 1E).

### 3.2. Metabolomic Analysis: Differentially Expressed Metabolites in the Two Varieties

To comprehensively understand the metabolic changes in the two varieties during fruit development, non-targeted metabolomics detection and analysis were performed on fruit flesh samples at 50, 80, 110, 140, and 170 DAFs. A principal component analysis (PCA) showed that PC1, PC2, and PC3 accounted for 26.53%, 25.58% and 11.93% of the variation in metabolite accumulation, respectively (Figure 2A). In the PCA score plot, the sample groups of different varieties and different periods are clearly separated, and the replicate samples were clustered together. A total of 1797 metabolites were identified in this study, with the top three categories being Lipids and lipid-like molecules (21.01%), Organic acids and derivatives (17.95%), and Phenylpropanoids and polyketides (17.45%, including 38.9% flavonoids) (Appendix A). Across five developmental stages, 503, 543, 624, 534, and 550 DEMs were observed between the two varieties (Appendix A). These DEMs were classified into 11 categories, with Phenylpropanoids and polyketides being the second most prevalent. Notably, in the five developmental stages, 44, 41, 46, 39, and 36 flavonoids within this category were differentially expressed in HF compared with LF, respectively (Figure 2B). Among these, Trilobatin, Cratenacin, and Betuletol 3-galactoside showed significant differences across all five stages (Figure 2C,D). To further investigate the content of flavonoids in the flesh of the two varieties, absolute quantitative analysis of flavonoid monomers was performed using fruit flesh samples collected at 170 DAFs. The results showed that the most abundant flavonoid monomer in the flesh was Procyanidin B2 (Flavonoid_145), followed by Chlorogenic acid (Flavonoid_263), Cryptochlorogenic acid (Flavonoid_244), and Sieboldin (Flavonoid_244). Notably, Sieboldin was present in higher amounts in HF but was undetectable in LF (Figure 2E).

### 3.3. Transcriptome Analysis

To further investigate the molecular mechanisms of the differences in metabolic accumulation between the two varieties, transcriptome sequencing analysis was performed on the flesh of the two pear varieties at five developmental stages. A total of 203.74 Gb clean reads were obtained after removing reads containing adapters and ploy-N and low-quality reads (Appendix A). More than 80.45% of clean reads in each library could be mapped to the pear reference genome. The percentages of Q30 and GC content in each library were higher than 95.02% and 46.82%, respectively, indicating that the transcriptomic data were of high quality and could be used for further DEG analysis (Appendix A). The Pearson correlation coefficient (PCC) revealed suitable intra-group biological repeatability in the samples of each stage. A total of 45,408 unigenes, of which 32,585 were annotated, and 21,896 were classified into different families.

To elucidate the patterns of gene expression changes across different developmental stages in the two varieties, K-means clustering analysis was performed on all genes, which were grouped into 9 distinct clusters according to their expression profiles (Appendix A). Among these clusters, the gene expression trends in Cluster 1 in HF and Cluster 3 in LF were consistent with the accumulation trends of flavonoids (Figure 3A). These two clusters contained 3228 and 7094 genes, respectively, with diverse expression patterns. The expression patterns of these genes likely indicate their involvement in the complex regulatory networks that modulate the overall response to flavonoid accumulation in the fruit flesh. Among these genes, a total of 494, 658, 697, 613, and 933 genes were differentially expressed between the two pear varieties at successive time points.

To identify the specific genes that contribute significantly to these patterns and potentially underlie the observed differences between the two varieties, we focused on identifying differentially expressed genes (DEGs). DEGs were screened using the criteria of |log2(FoldChange)| ≧ 1 and padj ≦ 0.05. As the fruit developed, starting from 50 DAFs, a total of DEGs with upregulated transcription levels in HF compared to LF were 3679, 4074, 2607, 3417, and 5547, respectively. Meanwhile, DEGs with downregulated transcription levels are 2888, 2657, 2037, 2352, and 4426 (Figure 3B). Among these differentially expressed genes, 1258 genes were differentially expressed across all five periods. A total of 33 transcription factors were screened from them, including 2 *MYB* transcription factors (*MYB4* and *MYB15*), 3 *ERF* transcription factors (*ERF5*, *ERF17*, and *ERF4*), 3 *WRKY* transcription factors (*WRKY70*, *WRKY50*, and *WRKY39*), 1 *bZIP* transcription factor, and 6 *bHLH* transcription factors. To further reveal the biological functions of DEGs in these samples, the KEGG enrichment analysis of DEGs was carried out. The results showed that all DEGs were successfully assigned to 131 KEGG pathways. In particular, the pathways of ‘Phenylpropanoid biosynthesis’ and ‘Flavonoid biosynthesis’ were significantly enriched across five and four developmental stages of the fruit, respectively. These results indicated that the DEGs related to the above secondary metabolite biosynthesis pathways might underlie the regulatory mechanisms contributing to the differences between the two varieties.

### 3.4. Weighted Gene Co-Expression Network Analysis

In this study, the weighted gene co-expression network (WGCNA) package tool was used to construct co-expression modules, and samples and genes with similar expression patterns were clustered to analyze the association between different modules and flavonoid-related phenotypes. Based on the above results, the total flavonoid content in HF fruits was significantly higher than that in LF fruits. To explore the molecular mechanisms underlying the varietal differences, ten metabolites classified as flavonoids were selected for further analysis. Specifically, these ten flavonoids showed the most significant differences between the two varieties at 110 days after flowering (DAFs), including morin (Com_596_pos), kaempferol 7-O-glucoside (Com_2987_pos), 3′,5′-Dihydroxy-3,5,6,7,8,4′-hexamethoxyflavone (Com_2959_pos), geraldol (Com_2265_pos), marein (Com_936_pos), choerospondin (Com_238_neg), taxifolin 7-O-beta-D-glucoside (Com_258_neg), isoquercetin (Com_255_neg), eriodictyol-7-O-glucoside (Com_249_neg), Okanin 4-methyl ether 3’-glucoside (Com_1065_neg). In this study, after merging highly similar modules, a total of 17 modules were identified (Figure 4A). Among these, the red module contained 658 genes and exhibited significant correlation with the target phenotypic traits. KEGG enrichment analysis revealed that genes in the red module were significantly enriched in the flavonoid biosynthesis pathway, indicating that these genes may play a crucial role in the phenotypic variations in flavonoid accumulation between the two varieties (Figure 4B). Therefore, seven structural genes were selected from the red module as key candidate genes involved in flavonoid biosynthesis, including *Pspp_Chr04_00864* (Phenylalanine ammonia-lyase, PAL), *Pspp_Chr01_01394* (Chalcone-flavanone isomerase, CHI), *Pspp_Chr15_02023* (Naringenin,2-oxoglutarate 3-dioxygenase, FHT), *Pspp_Chr15_03047* (Flavonol synthase, FLS), *Pspp_Chr15_00198* (dihydroflavonol 4-reductase, DFR), *Pspp_Chr06_00719* (Leucoanthocyanidin dioxygenase, ANS), *Pspp_Chr10_02574* (Anthocyanidin reductase, ANR) to conduct a co-expression network diagram. For network construction, edges with a correlation weight > 0.2 within the red module were filtered, resulting in the identification of 38 nodes. These nodes were visualized using Cytoscape software to generate the network diagram (Figure 4C). The transcriptional profiles of these nodes across five developmental stages in both varieties were depicted in a heatmap (Figure 4D). Through detailed analysis, we found that 10 structural genes, 1 gene annotated as an MYB transcription factor (MYB4), and 4 genes involved in plant hormone metabolism formed a co-expression network with the 7 structural genes related to flavonoid biosynthesis that we focused on.

### 3.5. RT-qPCR Validation

To validate the reliability of transcriptome data and confirm the expression levels of flavonoid biosynthesis-related genes, ten genes including *PAL* (*Pspp_Chr04_00864*), *4CL* (*Pspp_Chr01_01925*), *CHI* (*Pspp_Chr01_01394*), *CHS* (*Pspp_Chr13_02509*), *F3’H* (*Novel.1226*), *FLS* (*Pspp_Chr15_03047*), *FHT* (*Pspp_Chr15_02023*), *DFR* (*Pspp_Chr15_00198*), *ANS* (*Pspp_Chr06_00719*), and *LAR* (*Pspp_Chr13_00394*) were selected, as shown in Figure 5. RT-qPCR detection was performed using the pear Histone gene as an internal reference gene. The results showed that these genes exhibited similar expression trends in both RT-qPCR and RNA-seq technologies, indicating that the RNA-seq data were authentic and reliable. In the HF variety, except for the *LAR* gene, the other nine genes showed high expression levels in fruits from 50 to 140 days after flowering (DAFs), and the expression levels dropped sharply at 170 DAFs. In the LF variety, the nine genes showed high expression levels at 80 DAFs and 110 DAFs, decreased at 140 DAFs, and increased at 170 DAFs, especially for the *4CL* and *LAR* genes. In the comparison between HF and LF varieties, from 50 to 140 DAFs, the expression levels of the nine genes (except LAR) in the HF variety were higher than those in the LF variety; at 170 DAFs, the expression levels of all ten genes in the LF variety were higher than those in the HF variety. In conclusion, the expression levels and trends of flavonoid biosynthesis genes differed significantly between the HF and LF varieties, leading to significant differences in flavonoid content between the two varieties.

## 4. Discussion

### 4.1. Metabolic Profiling: The Composition, Concentration and Accumulation Patterns of Flavonoids Vary Among Different Pear Varieties

Pear fruit quality is affected by many factors, including the accumulation of metabolites such as reducing sugars, titratable acids, amino acids and total flavonoids [19]. Flavonoids, as one of the most important secondary metabolites in pear fruit, play a crucial role in the flavor and nutritional value of the fruit, but research on their accumulation and metabolic in pear flesh remains limited.

This study focused on analyzing the differences in flavonoid accumulation in flesh between two pear varieties with similar appearances. Firstly, the results showed that the accumulation of total flavonoids in the two varieties showed significant differences at each developmental stage (Figure 1D), with proanthocyanidin B, chlorogenic acid, and neochlorogenic acid being the predominant flavonoid compounds in the flesh of pear fruits during the ripening period (Figure 2E). This finding contrasts with a previous study [20], which found epicatechin and rutin to be the predominant flavonoids in the peel and in the flesh of ten pear varieties (*Pyrus* spp.). Besides, we found that the content of trilobatin in the HF flesh was consistently and significantly higher than that in the LF flesh during fruit development. Trilobatin, a natural dihydrochalcone compound, is primarily found in sweet tea (*Lithocarpus polystachyus* Rehd.) and certain Malus species. In recent years, due to its remarkable activities such as antioxidant [21], anti-inflammatory [22], hypoglycemic [23], and neuroprotective effects [24], trilobatin has garnered extensive attention in the fields of food additives, drug development, and metabolic engineering. The findings indicate that there are significant differences in both the total amount and types of flavonoids among different pear varieties. Specifically, the flesh of the HF variety is rich in flavonoids, which may suggest potential applications in the food industry as raw materials for obtaining products of high nutritional quality and perceived health benefits, or in the pharmaceutical industry for their medicinal value.

Cho, et al. [25] reported that in 7 pear varieties, the levels of vitamins (ascorbic acid and alpha tocopherol), arbutin, chlorogenic acid, malaxinic acid, total caffeic acid, total flavonoids, and total phenolics were highest in ripe pear fruit 20 DAFs and then gradually decreased. In the present study, similar trends were observed in the flesh of the LF, with these compounds reaching their highest levels early in development and then gradually declining. In contrast, the contents of these compounds in the HF variety peaked 110 days after ripening and then declined. This suggests that the accumulation pattern of flavonoids in HF fruit differs from that of most varieties, possibly related to their respective accumulation mechanisms.

Flavonoids, a diverse group of secondary metabolites, are synthesized via the phenylpropanoid pathway, which converts phenylalanine into cinnamic acid through the action of phenylalanine ammonia-lyase (PAL), and then transforms cinnamic acid into various phenylpropanoid intermediates that are further modified in the flavonoid biosynthetic pathway to produce a wide array of flavonoid compounds [26]. As the precursor of flavonoids, phenylalanine is commonly presumed to have a positive correlation with flavonoid content [27]. However, in the present study, the phenylalanine concentration in HF flesh was found to be significantly lower than that in LF at all developmental stages. This paradox is similar to the results of Dai, et al. [28] on anthocyanins in grape berries. A possible explanation for this negative correlation between phenylalanine and flavonoids concentrations might be that the metabolic flux through L-phenylalanine ammonia lyase (PAL) is higher than in HF flesh than in LF. This increased flux could lead to a more rapid conversion of phenylalanine into downstream metabolites, thereby reducing the pool of free phenylalanine available for flavonoid synthesis. Soubeyrand, et al. [29] reported that the anthocyanin content increased in grape cells cultured under high C/N ratio conditions, with a 38% increase in PAL metabolic flux, which supports this hypothesis. Additionally, the concentration of 4-coumarate in the flesh of the HF variety was significantly higher than that in the LF variety at 110 and 170 DAFs. To elucidate the mechanisms underlying the differences in flavonoid accumulation between the flesh of HF and LF, further investigations into the expression of related genes and enzyme activities are required.

### 4.2. Transcriptomic Insights into the Molecular Mechanisms Underlying Differences in Primary and Secondary Metabolite Accumulation Between the Two Pear Varieties

To further elucidate the mechanisms underlying the differences in flavonoid accumulation between the two varieties, we conducted a detailed analysis of the transcriptomes across various developmental stages. First, we applied K-means clustering on the transcriptomic data from these stages and found that several genes involved in sugar metabolism or transport clustered in clusters with similar flavonoid accumulation patterns. These genes included *Pspp_Chr17_02347* (*SUS*), *Pspp_Chr17_02348* (*SUS*), *Pspp_Chr05_00022* (*SPSA*), *Pspp_Chr02_00200* (*SPSA*), *Pspp_Chr05_00583* (*GLGL1*), *Pspp_Chr10_02629* (*GPT*), *Pspp_Chr13_01001* (*G6PDC*), *Pspp_Chr09_00853* (*PFPB*) and *Pspp_Chr10_02233* (*SWEET*), and these genes were differentially expressed between the two varieties. Alterations in gene expression during various stages of fruit development could significantly affect metabolic processes, among which the accumulation of secondary metabolites is often affected by both the content of primary metabolites and the expression of genes involved in biosynthesis pathways [30,31]. Studies have shown a significant correlation between sugar content, the expression of sugar metabolism-related genes, and the accumulation of flavonoids [32,33,34]. For instance, the accumulation of sugar and total flavonoids in *Rosa roxburghii* fruit showed a significant positive correlation, and also positively related to the activities of enzymes such as SS, SPS and IVR [35]. Light exposure can regulate flavonoid synthesis by modulating the expression of genes such as *SWEET1* and *ATHB12*, thereby affecting the flavonoid content in plant [36]. Therefore, the differential expression of these sugar metabolism-related genes in the two pear varieties may contribute to the observed differences in flavonoid accumulation, thereby highlighting the crosstalk of primary and secondary metabolic pathways in regulating fruit quality.

Moreover, the expression of structural genes involved in the flavonoid biosynthesis pathway often directly regulates the levels of various flavonoids [37]. Previous studies on genes involved in flavonoid biosynthesis in pear, which employed comparative genomic analysis and enzymatic characterization with apple as a reference, have elucidated the relationship between these genes and the characteristic flavonoid metabolite profiles in pear [2]. In this study, K-means cluster analysis revealed that the expression patterns of genes such as *Pspp_Chr15_03047* (annotated as flavanone 3-hydroxylase) and *Pspp_Chr03_02092* (annotated as Chalcone-flavanone isomerase) were similar to the total flavonoid content and displayed differential expression between the two varieties, which suggested that the accumulation of flavonoids in pear fruit and the differences between varieties are regulated by related synthesis genes. Meanwhile, KEGG enrichment analysis on the transcriptome showed that DEGs were significantly enriched in the “phenylpropanoid biosynthesis” and “flavonoid biosynthesis” pathways across various developmental stage. RT-qPCR analysis of key structural genes in the flavonoid biosynthesis pathway confirmed that nine structural genes were highly expressed in HF. Among them, the expression patterns of *PAL* (*Pspp_Chr04_00864*), *CHI* (*Pspp_Chr01_01394*), *F3′H* (*Novel.1226*), *FLS* (*Pspp_Chr15_03047*), and *ANS* (*Pspp_chr06_00719*) were found to be similar to the accumulation patterns of total flavonoids in the two varieties. Taken together, these findings suggest that the accumulation of flavonoids in the pear varieties is closely related to the expression of these structural genes.

A substantial body of research has demonstrated that transcription factors (TFs) play crucial roles in the regulation of the accumulation of flavonoids [19,38,39]. For example, Rui et al. found that *PbMYB10b* and *PbMYB9* genes were closely related to the synthesis pattern of flavonoids in pear fruits, with *PbMYB10b* regulating anthocyanin and proanthocyanidin pathways via *PbDFR* induction, while *PbMYB9* not only activates the proanthocyanidin pathway by specifically targeting the *PbANR* promoter but also induces anthocyanin and flavonol synthesis by binding to the *PbUFGT1* promoter [40]. Moreover, it has been reported that transcription factors from the *ERF* [14,41,42], *NAC* [43], as well as from the *WRKY* [44,45] and *bZIP* [19] families, were also involved in the regulation of flavonoid biosynthesis. In this study, we identified a total of 33 transcription factors among 1258 genes that showed differential expression across all developmental stages, including 2 *MYB*, 3 *ERF*, 3 *WRKY*, 1 *bZIP*, and 6 *bHLH* transcription factors. These transcription factors might be involved in the regulation of flavonoid biosynthesis genes, leading to the differences in flavonoid accumulation between the two varieties.

Differences in flavonoid accumulation between pear varieties are often affected by multiple factors, and the molecular mechanisms behind them are also complex [37,46]. To elucidate these mechanisms, we conducted a weighted gene co-expression network analysis (WGCNA) and constructed a co-expression network comprising key structural genes in the flavonoid biosynthesis pathway (Figure 4B). The co-expression network included 10 structural genes, of which nine were directly involved in flavonoid biosynthesis, namely *Pspp_Chr05_02768* (annotated as Anthocyanidin reductase ((2S)-flavan-3-ol-forming) [47], *Pspp_Chr06_00295* (annotated as cinnamoyl-CoA reductase-like), *novel.2759* (Flavonoid 3′-monooxygenase), *Pspp_Chr13_02511* (Chalcone and stilbene synthases), *Pspp_Chr07_01837* (Chalcone-flavanone isomerase), *Pspp_Chr04_00024* (Chalcone and stilbene synthases), *novel.1171* (Flavonoid 3′-monooxygenase), *Pspp_Chr01_01088* (Chalcone--flavanone isomerase), and *novel.1105* (Anthocyanidin reductase). These genes play crucial roles in various steps of the flavonoid biosynthesis pathway, from the initial synthesis of flavonoid precursors to the final modification of flavonoid compounds [2,10,47]. In addition to structural genes, a transcription factor annotated as *MYB4* was identified in the co-expression network. This *MYB4* transcription factor was co-expressed with multiple structural genes, suggesting its potential regulatory role in flavonoid biosynthesis. *MYB* transcription factors are well-documented regulators of the flavonoid biosynthesis pathway. Liu, et al. [48] confirmed identified *MYB4-like1* and *MYB4-like2* as key regulators of anthocyanin biosynthesis in ‘pingguo’ pear fruit peels through transcriptome and qPCR data analysis. *PbMYB12b* has been shown to promote the expression of chalcone synthase (CHS) and flavonol synthase (FLS), thereby enhancing the biosynthesis of flavonoids [49]. Therefore, combined with our findings, further studies are needed to elucidate the specific roles of these genes and their interactions in different pear varieties.

## 5. Conclusions

Despite the crucial importance of flavonoids, research on their accumulation patterns and molecular mechanisms in pear flesh is limited. This study provides valuable insights into the molecular mechanisms of flavonoid accumulation in pear fruit flesh. By conducting comprehensive, multi-time-point metabolomic and transcriptomic analysis of two pear varieties, HL and LF, which exhibit significant differences in total flavonoid content, we successfully constructed a dynamic profile of flavonoid metabolite accumulation in pear flesh. Using WGCNA, we identified key structural genes and candidate transcription factors that are highly correlated with flavonoid synthesis. One of these transcription factors, *MYB4*, warrants further investigation. Furthermore, genes associated with auxin response and jasmonic acid metabolism also showed high correlations with the differential flavonoid content between the two varieties. This suggests that these hormonal pathways may play an important role in modulating flavonoid accumulation in pear fruit flesh.

In summary, this study advances our understanding of the genetic and metabolic basis of fruit quality and provides a theoretical foundation for future breeding programs aimed at improving pear fruit quality and nutritional value.

## Figures and Tables

**Figure 1 foods-14-03716-f001:**
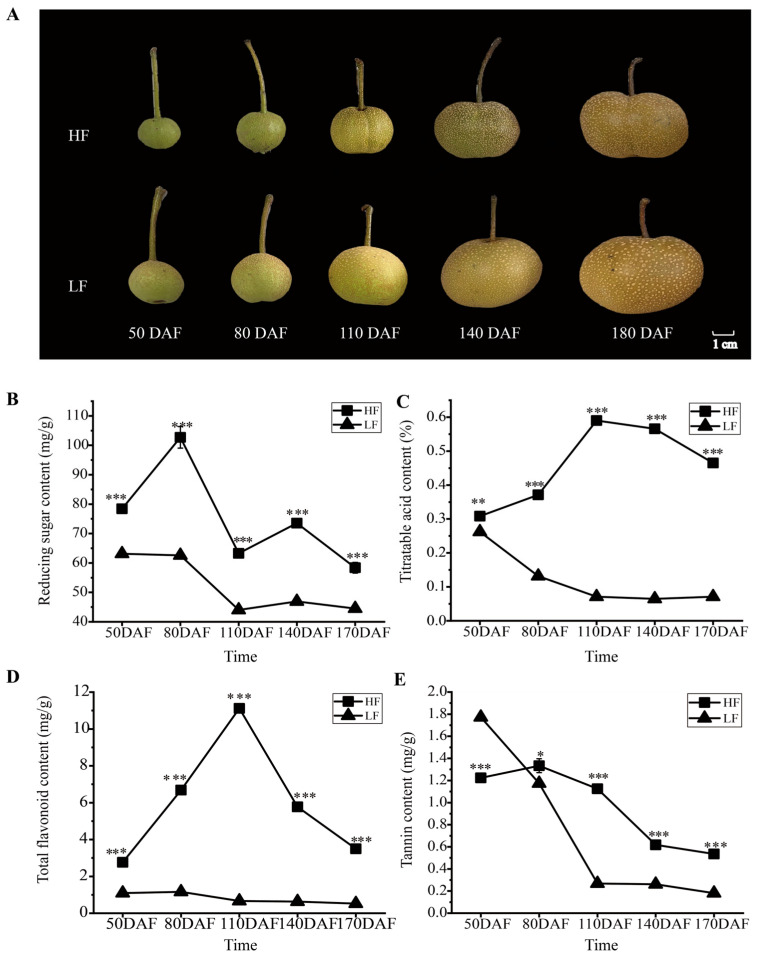
Phenotypic and physiological changes in HF and LF at different developmental stages. (**A**): Phenotypes of fruits of the two pear varieties at different periods after flowering. HF and LF represent ‘Heqingxiaoshali’ and ‘Lunanhuangpili’, respectively; DAFs indicates days after flowering. (**B**): Reducing sugar content in the flesh of the two varieties. (**C**): Titratable acid content in the flesh of the two varieties. (**D**): Total flavonoid content in the flesh of the two varieties. (**E**): Tannin content in the flesh of the two varieties. Vertical bars indicate SE (*n* = 3). * represents *p* < 0.05; ** represents *p* < 0.01; and *** represents *p* < 0.001.

**Figure 2 foods-14-03716-f002:**
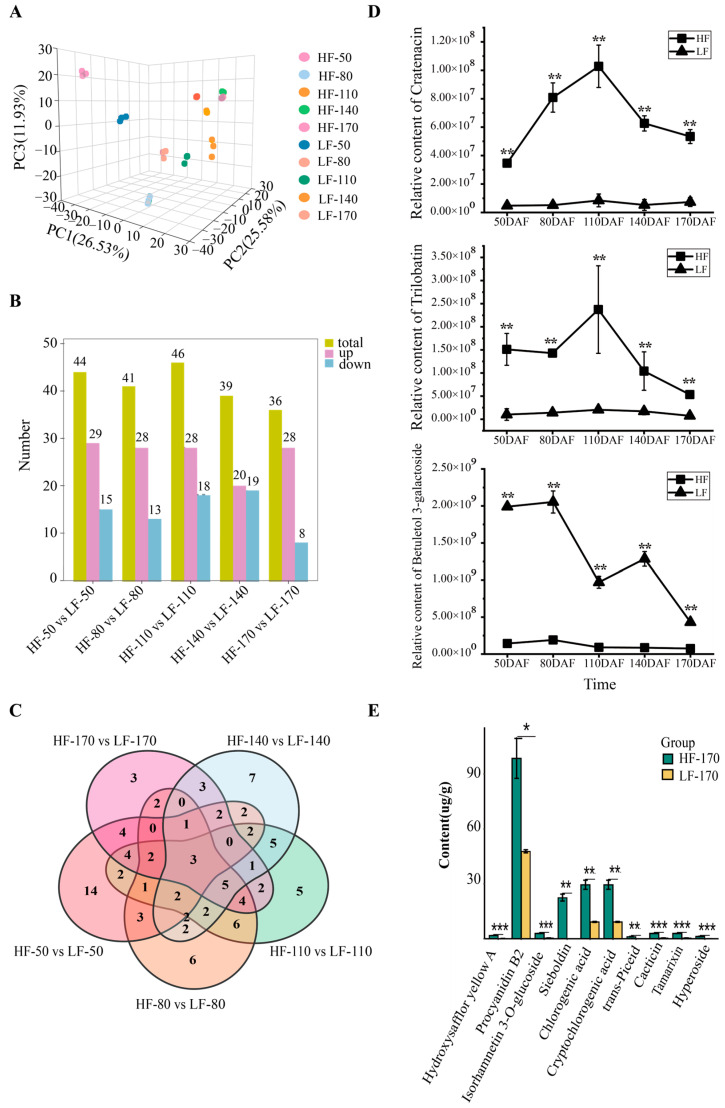
Metabolomic analysis of HF and LF at different developmental stages. (**A**): Principal component analysis of flesh samples of the two pear varieties at different periods. The labels HF and LF represent ‘Heqingxiaoshali’ and ‘Lunanhuangpili’, respectively. The numbers in the middle represent 50, 80, 110, 140, and 170 DAFs, and the numbers after the “-” represent replicates. (**B**): Statistical analysis of DEMs. The yellow module represents the total number of DEMs compared between the two varieties, the pink module represents the number of metabolites upregulated in ‘Heqingxiaoshali’ compared to ‘Lunanhuangpili’, and the blue module represents the number of metabolites downregulated in ‘Heqingxiaoshali’ compared to ‘Lunanhuangpili’. (**C**): Venn diagram of differential metabolites compared between the two varieties at different stages, the number in the circles or areas of different sets represent the number of metabolites contained in the area. (**D**): The concentrations of three metabolites showed significant differences in all five stages between the two varieties. (**E**): Absolute quantification of flavonoid metabolites in two pear fruits at 170 DAFs. Vertical bars indicate SE (*n* = 3). * represents *p* < 0.05; ** represents *p* < 0.01; and *** represents *p* < 0.001.

**Figure 3 foods-14-03716-f003:**
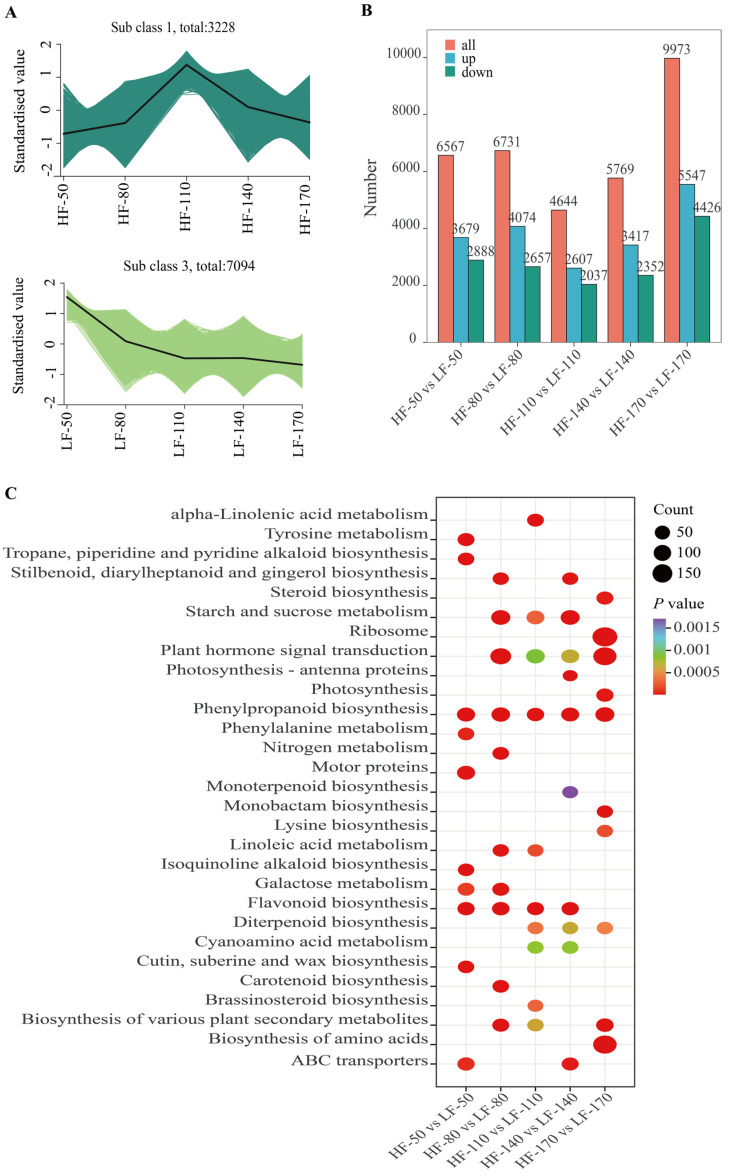
Transcriptomic analysis of the flesh of two pear varieties at five developmental stages. (**A**): Results of K-means cluster analysis of time series of the pulp for the two pear varieties. HF and LF represent ‘Heqingxiaoshali’ and ‘Lunanhuangpili’, respectively. The numbers after “-” indicate 50, 80, 110, 140, and 170 DAFs. (**B**): Statistical analysis of DEGs at different periods after flowering in the two pear varieties. The red module represents the total number of DEGs, the blue module represents the number of differentially expressed genes in ‘Heqingxiaoshali’ compared to ‘Lunanhuangpili’, and the green module represents the number of down-regulated differentially expressed genes in ‘Heqingxiaoshali’ compared to ‘Lunanhuangpili’. (**C**): KEGG pathway enrichment analysis of all DEGs. Circle size represents the number of enrichments, and color represents the significance of enrichment. Darker colors indicate more significant enrichment.

**Figure 4 foods-14-03716-f004:**
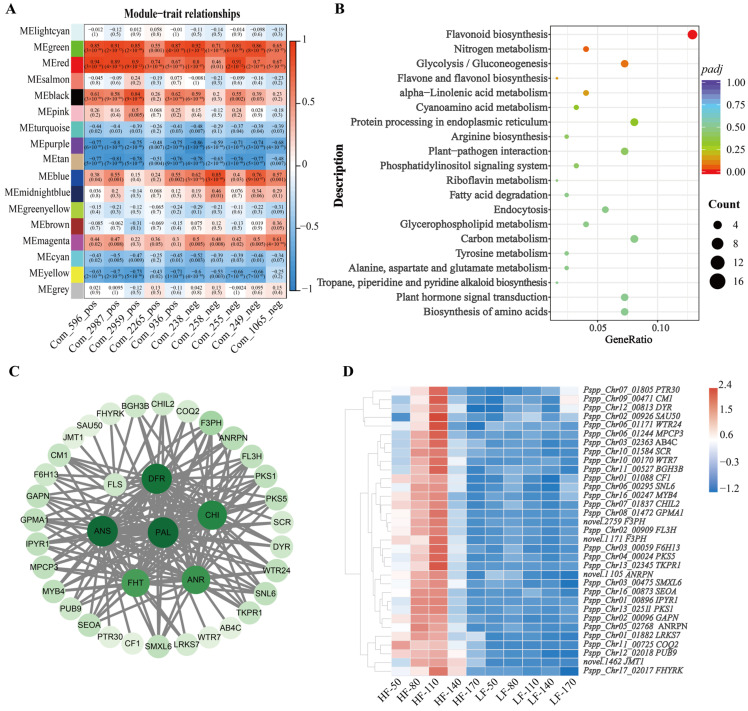
WGCNA of genes in HF and LF. (**A**): Module-trait correlation plot, with each row corresponding to a module and each column corresponding to a trait. Each cell contains the corresponding correlation and *p* value. The table is color-coded according to the color legend based on correlation. (**B**): KEGG pathway enrichment analysis of genes in red module. (**C**): The correlation networks of genes in the ‘red’ module, in which connected with 7 structural genes (*PAL*, *CHI*, *DFR*, *ANS*, *ANR*, *FHT* and *FLS*) ≥ 0.2 are displayed, and the larger the circle, the higher the connectivity. (**D**): Heatmap of expression of 33 nodes associated with structural genes in Figure 4C.

**Figure 5 foods-14-03716-f005:**
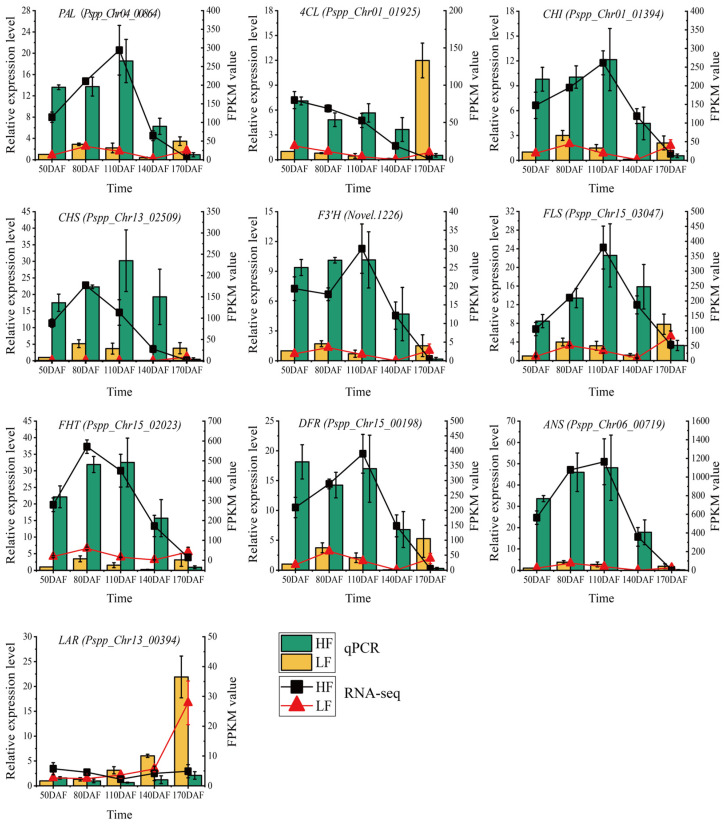
qRT-PCR validation of structural genes in the phenylpropanoid biosynthesis pathway. The bar chart represents qRT-PCR results, and the line chart represents RNA-seq data. Vertical bars indicate SE (n = 3).

## Data Availability

The original contributions presented in this study are included in the article/Appendix A. Further inquiries can be directed to the corresponding authors.

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
