# Peer review of "Integrated Analysis of Metabolome and Transcriptome Provides Insights into Flavonoid Biosynthesis of Pear Flesh (*Pyrus pyrifolia*)"

_foods, 2025, doi:10.3390/foods14213716_

Round 1
Reviewer 1 Report
Comments and Suggestions for Authors
The study provides valuable insights into the molecular mechanisms of flavonoid accumulation in pear flesh using an integrated metabolomic and transcriptomic approach, and the biological interpretation of the WGCNA results is compelling. However, the methodology described for the non-targeted metabolomics analysis lacks several critical details necessary to ensure the validity and quantitative accuracy of the reported metabolite data.
We recommend that the authors address the following points regarding Section 2.3 (Metabolite Sequencing and Analysis):
- Metabolite Identification Confidence
The methodology outlines the use of LC-MS/MS and XCMS software for data processing. However, there is a lack of information regarding how the metabolites were identified and the degree of confidence associated with these identifications.
- Please elaborate on the criteria used for metabolite identification (e.g., specific databases utilized, matching scores, confirmation via MS/MS fragmentation patterns, or use of chemical standards). Providing this detail is crucial, especially when discussing essential compounds like Trilobatin, Cratenacin, and Betuletol 3-galactoside that showed significant differences across all stages.
- Validation of Quantitative Accuracy and Linearity
The majority of the analysis relies on relative quantification (peak area), though absolute quantitative analysis of flavonoid monomers was performed on 170 DAF samples.
- For the absolute quantification of monomers (e.g., Procyanidin B2), please provide evidence of quantitative accuracy, including the linearity (standard curves and R² values) established between the signal intensity and the actual concentration.
- Even for relative quantification, describing quality metrics that confirm the instrument's dynamic range and signal reproducibility for diverse compound classes is highly recommended.
- Handling of Batch Effects and Analytical Reproducibility
The study involved 30 samples collected across five different developmental time points.
- Please clarify whether the LC-MS/MS analyses were performed in a single analytical batch or multiple batches. If numerous batches were used, the manuscript must detail the specific strategies employed to normalize the data and correct for potential batch effects, instrumental drift, or run-to-run variability, beyond simply retaining metabolites with a CV < 30% in the QC samples.
- Assurance of Data Stability Across Time Points
Samples were collected over a long period (March to August 2024) and immediately flash-frozen and stored at -80℃.
- While immediate freezing is standard practice, the overall stability and reproducibility of the quantitative data across the entire analytical sequence are vital for a time-series study. Please provide further assurance that any potential time-dependent degradation or analytical shifts did not confound the biological differences observed between the five developmental stages. The clustering of biological replicates in the PCA plot suggests good intra-group consistency. Still, clarity on the QC strategy designed to monitor inter-batch or long-term analytical stability should be added.
Author Response
Comments 1: The study provides valuable insights into the molecular mechanisms of flavonoid accumulation in pear flesh using an integrated metabolomic and transcriptomic approach, and the biological interpretation of the WGCNA results is compelling. However, the methodology described for the non-targeted metabolomics analysis lacks several critical details necessary to ensure the validity and quantitative accuracy of the reported metabolite data.
Response 1: Thank you very much for your positive evaluation of our research. In response to your suggestions for revision, we have made corresponding additions and corrections in the main text. Thank you again for taking the time to review our manuscript.
Comments 2: Please elaborate on the criteria used for metabolite identification (e.g., specific databases utilized, matching scores, confirmation via MS/MS fragmentation patterns, or use of chemical standards). Providing this detail is crucial, especially when discussing essential compounds like Trilobatin, Cratenacin, and Betuletol 3-galactoside that showed significant differences across all stages.
Response 2: In the present study, metabolite identification was conducted using a secondary spectral library that integrates authentic standards and secondary spectra sourced from multiple public databases, including HMDB, RefMetaDB, and ReSpect for Phytochemicals. The reliability of metabolite identification was assessed on a five-tier scale, ranging from level 0 (highest reliability) to level 4 (lowest reliability), as described in Metabolites (2018). Specifically, in our study, 290 metabolites were identified at level 1, 859 at level 2, and 648 at level 3. These details have now been incorporated into the “Materials and Methods” section of the manuscript (Line 152-157).
Blaženović, I.; Kind, T.; Ji, J.; Fiehn, O. Software tools and approaches for compound identification of LC-MS/MS data in metabolomics. Metabolites 2018, 8, 31, doi:org/10.3390/metabo8020031.
Comments 3: The majority of the analysis relies on relative quantification (peak area), though absolute quantitative analysis of flavonoid monomers was performed on 170 DAF samples.
Response 3: Agree, our analysis primarily relied on relative quantification. Absolute quantification was employed for two main purposes: first, to validate the reliability of the relative quantitative results, and second, to gain a deeper understanding of the composition of flavonoids in the flesh of mature pears, as well as the differences between the two varieties.
Comments 4: For the absolute quantification of monomers (e.g., Procyanidin B2), please provide evidence of quantitative accuracy, including the linearity (standard curves and R² values) established between the signal intensity and the actual concentration.
Response 4: To quantitatively analyze flavonoids, we first prepared standard solutions of 10 kinds of flavonoids: 0.5 nmol/L, 1 nmol/L, 5 nmol/L, 10 nmol/L, 20 nmol/L, 50 nmol/L, 100 nmol/L, 200 nmol/L, 500 nmol/L, 1000 nmol/L, 2000 nmol/L, 5000 nmol/L, 10,000 nmol/L, and 20,000 nmol/L. The chromatographic peak intensities corresponding to the quantitative signals at each standard concentration were measured. A calibration curve was then plotted with the external standard to internal standard concentration ratio (or external standard concentration) on the horizontal axis and the external standard to internal standard peak area ratio (or external standard peak area) on the vertical axis. Finally, the integrated peak area ratios for all tested samples were substituted into the linear equation of the calibration curve to calculate the concentrations. Further calculations using appropriate formulas yielded the actual concentrations of the compounds in the samples. This detailed method has been added to the “Materials and Methods” section of this manuscript (Line 169-182) and is included in Supplementary Table 1.
Comments 5: Even for relative quantification, describing quality metrics that confirm the instrument's dynamic range and signal reproducibility for diverse compound classes is highly recommended.
Response 5: Thanks for your suggestion. For the quality control indicators for confirming the instrument's dynamic range and the signal reproducibility of different compound categories, we mainly determined them in the following two aspects: First, during the instrument analysis process, quality control (QC) samples were inserted into the analytical samples. By overlapping and analyzing the total ion current (TIC) chromatograms of the same QC sample obtained from mass spectrometry detection, we assessed the instrument's stability throughout the project. The overlapped TICs are shown in the figure below:
Second, the stability of the instrument detection was judged through the distribution of the coefficient of variation (CV) values of the QC samples. The empirical cumulative distribution function can be used to analyze the frequency of substances with CV values lower than the reference value. A higher proportion of substances with low CV values in the QC samples indicates more stable experimental data: if more than 80% of the substances have CV values less than 0.3, the experimental data are considered stable; if more than 80% of the substances have CV values less than 0.2, the experimental data are considered very stable. The distribution of CV values for the two groups of samples and QC samples in this experiment is shown in the figure below. As can be seen from the figure, the proportion of substances with CV values less than 0.2 in the QC samples is over 80%, indicating very stable data.

Comments 6: The study involved 30 samples collected across five different developmental time points. Please clarify whether the LC-MS/MS analyses were performed in a single analytical batch or multiple batches. If numerous batches were used, the manuscript must detail the specific strategies employed to normalize the data and correct for potential batch effects, instrumental drift, or run-to-run variability, beyond simply retaining metabolites with a CV < 30% in the QC samples.
Response 6: Thank you for pointing this out. All samples were tested and analyzed in the same batch, thus eliminating batch effects. Details of sampling and sample storage were provided in the “Materials and Methods” section of this manuscript (Line 119-122). The description is as follows: “Sampling was carried out on a clear morning. After the samples for all time periods were collected, they were tested uniformly. The samples were immediately stored in liquid nitrogen after collection and then stored in a -80℃ refrigerator after returning to the laboratory.”
Comments 7:Samples were collected over a long period (March to August 2024) and immediately flash-frozen and stored at -80℃.While immediate freezing is standard practice, the overall stability and reproducibility of the quantitative data across the entire analytical sequence are vital for a time-series study. Please provide further assurance that any potential time-dependent degradation or analytical shifts did not confound the biological differences observed between the five developmental stages. The clustering of biological replicates in the PCA plot suggests good intra-group consistency. Still, clarity on the QC strategy designed to monitor inter-batch or long-term analytical stability should be added.
Response 7: We totally agree with your point. In this experiment, we utilized QC samples for quality control purposes. The first three QC samples were introduced prior to injection to monitor the instrument's status before injection and to equilibrate the chromatography-mass spectrometry system. The subsequent three QC samples were subjected to segmented scanning, and the resulting MS/MS spectra, along with those obtained from the experimental samples, were employed for metabolite identification. The QC samples interspersed throughout the sample analysis were used to assess the system's stability throughout the entire experimental process and to conduct data quality control analysis.
Reviewer 2 Report
Comments and Suggestions for Authors
In the Material and Methods section, the number of samples are needed.
Because your PCA (Fig 2) plot shows that the first two components only represent around 61% of total variance, I would recommend using 3D PCA score plot to present the sample distribution under the first three PC components.
Did you discuss why the CV of trilobatin content under 110 DAF (HF) is higher that others? The analytical error or other things?
Author Response
Thank you very much for taking the time to review this manuscript. Please find detailed responses below and the corresponding revisions/corrections highlighted/in track changes in the re-submitted files.
Comments 1: In the Material and Methods section, the number of samples are needed. Because your PCA (Fig 2) plot shows that the first two components only represent around 61% of total variance, I would recommend using 3D PCA score plot to present the sample distribution under the first three PC components.
Response 1: Thank you for your suggestion, we have made corresponding changes in the manuscript, as shown in Figure 2 (Line 266).
Comments 2: Did you discuss why the CV of trilobatin content under 110 DAF (HF) is higher that others? The analytical error or other things?
Response 2: Thank you for raising this question. We have carefully examined this observation and would like to provide the following explanation: The higher CV observed for trilobatin content under the 110 DAF (HF) condition may primarily be attributed to inherent biological variability within the samples. At this specific developmental stage, the metabolic pathways responsible for trilobatin synthesis could be more susceptible to fluctuations due to genetic, environmental, or physiological factors. For instance, slight differences in fruit maturity, microenvironmental conditions (e.g., light exposure, temperature), or individual plant genetic variations could lead to more pronounced variations in trilobatin levels compared to other time points or conditions. This may indicate greater changes in pear pulp metabolism during this period.
Round 2
Reviewer 2 Report
Comments and Suggestions for Authors
The discussion regarding the 3D PCA plot is missing.
Author Response
Thank you very much for taking the time to review this manuscript again. Please find the detailed responses below, with the corresponding revisions highlighted in the resubmitted files.
Comments 1: The discussion regarding the 3D PCA plot is missing.
Response 1: Thank you for your kind reminder. We have made the corresponding changes in the manuscript, as shown in Lines 235-237. In addition, we have corrected some formatting issues in the article, such as lines 266-267, lines 382-383, and line 402.